# *Acinetobacter baumannii* and Cefiderocol, between Cidality and Adaptability

Stefano Stracquadanio,[a] Carmelo Bonomo,[a] Andrea Marino,[b] Dafne Bongiorno,[a] Grete Francesca Privitera,[c] Dalida Angela Bivona,[a] Alessia Mirabile,[a] Paolo Giuseppe Bonacci,[a] Stefania Stefani[a]

[a]Biomedical and Biotechnological Sciences Department, University of Catania, Catania, Italy
[b]Department of Clinical and Experimental Medicine, University of Catania, Catania, Italy
[c]Department of Clinical and Experimental Medicine, Bioinformatics Unit, University of Catania, Catania, Italy

**ABSTRACT**   Among the bacterial species included in the ESKAPE group, *Acinetobacter baumannii* is of great interest due to its intrinsic and acquired resistance to many antibiotics and its ability to infect different body regions. Cefiderocol (FDC) is a novel cephalosporin that is active against Gram-negative bacteria, with promising efficacy for *A. baumannii* infections, but some studies have reported therapeutic failures even in the presence of susceptible strains. This study aims to investigate the interactions between FDC and 10 *A. baumannii* strains with different susceptibilities to this drug. We confirmed diverse susceptibility profiles, with resistance values close to the EUCAST-proposed breakpoints. The minimal bactericidal concentration (MBC)/MIC ratios demonstrated bactericidal activity of the drug, with ratio values of $\leq 4$ for all of the strains except ATCC 19606; however, bacterial regrowth was evident after exposure to FDC, as were changes in the shapes of colonies and bacterial cells. A switch to a nonsusceptible phenotype in the presence of high FDC concentrations was found in 1 strain as an adaptation mechanism implemented to overcome the cidal activity of this antibiotic, which was confirmed by the presence of heteroresistant, unstable subpopulations in 8/10 samples. Genomic analyses revealed the presence of mutations in penicillin-binding protein 3 (PBP3) and TonB3 that were shared by all of the strains regardless of their resistance phenotype. Because our isolates harbored $\beta$-lactamase genes, $\beta$-lactamase inhibitors showed the ability to restore the antimicrobial activity of FDC despite the different nonsusceptibility levels of the tested strains. These *in vitro* results support the concept of using combination therapy to eliminate drug-adapted subpopulations and regain full FDC activity in this difficult-to-treat species.

**IMPORTANCE**   This work demonstrates the underrated presence of *Acinetobacter baumannii* heteroresistant subpopulations after exposure of *A. baumannii* strains to FDC and its instability. Both *A. baumannii* and FDC are of great interest for the scientific community, as well as for clinicians; the former represents a major threat to public health due to its resistance to antibiotics, with related costs of prolonged hospitalization, and the latter is a novel, promising cephalosporin currently under the magnifying glass.

**KEYWORDS**   *Acinetobacter baumannii*, cefiderocol, heteroresistance, AST, BLI, antimicrobial susceptibility testing, $\beta$-lactamases inhibitors

**A**cinetobacter baumannii is a major clinical threat, both because of its established role as an important pathogen associated with nosocomial infections at various body sites and because of the challenge posed by the very limited treatment options available, due to its high intrinsic and acquired antimicrobial resistance. The latter entails, in many cases and for different antibiotics, characteristics of instability, related to frequent genomic rearrangements involving gene amplification, mobile genetic element (MGE) acquisition, and resistome and phage profile changes, as recently demonstrated (1, 2).

Address correspondence to Stefania Stefani, stefania.stefani@unict.it.
The authors declare no conflict of interest.

The recent introduction of the novel siderophore-conjugated cephalosporin cefiderocol (FDC), whose spectrum includes *A. baumannii*, has raised great expectations for therapy (3). The molecule is composed of a siderophore component that binds iron and uses active iron transport for drug entry into the bacterial periplasmic space. The cephalosporin moiety is the active antimicrobial component, structurally resembling a hybrid between ceftazidime and cefepime. Like other $\beta$-lactam agents, the main bactericidal activity of FDC occurs through the inhibition of bacterial cell wall synthesis via binding of penicillin-binding proteins (PBPs) and inhibition of peptidoglycan synthesis, leading to cell death (4).

Several clinical studies pointed out that this drug had efficacy similar to that of the best therapies available for infections caused by Gram-negative bacteria, but a higher all-cause mortality rate was described for the subset of *A. baumannii* infections (5), despite a $MIC_{90}$ of 4 mg/L. Heteroresistance has been suggested as a possible reason for these failures (5, 6), especially if bacteria are classified as susceptible by standard antibiotic susceptibility testing (AST) (1).

Heteroresistance is a poorly understood mechanism of survival in the presence of antibiotics and is defined as the presence of subpopulations with MIC values higher (variably defined as $\geq$2- to 8-fold higher) than that in the main population, i.e., a susceptible bacterial isolate could harbor a minority of resistant subpopulations. Despite the discovery of a high prevalence of this mechanism in many different species (both Gram positive and Gram negative) with respect to many antibiotics (1), the majority of these subpopulations were considered unstable. Heteroresistance in *A. baumannii* has already been described for different classes of antibiotics, including colistin, aminoglycosides, imipenem, meropenem, and tigecycline (7–10).

Due to the problem of reproducibility of broth microdilution (BMD) testing (11), as well as reported errors in disk diffusion (DD) (12), we performed some cidality tests and population analysis profiling (PAP) on a subset of isolates representative of the entire sample with the aim of determining the best *in vitro* conditions to test FDC activity with *A. baumannii* strains with different degrees of susceptibility to this drug. Our results demonstrated that this species has a great potential for heteroresistance, despite an initial result of bactericidal activity of FDC obtained from the evaluation of the minimal bactericidal concentration (MBC)/MIC ratio. Furthermore, these subpopulations, which in some cases emerged upon exposure to FDC, were not stable at all, as they returned to the initial MIC after two passages in an antibiotic-free medium. Interestingly, the addition of avibactam and sulbactam eliminated all subpopulations.

## RESULTS

**Genome analyses.** The strains included in study are described in Table 1. All isolates were sequenced by next-generation sequencing (NGS) and belong to sequence type 2 (ST2) (7 strains), ST52 (2 strains), and ST77 (1 strain). All isolates contained several OXA gene alleles (Table 1), including chromosomal OXA-51-like genes (i.e., $OXA_{66}$, $OXA_{82}$, $OXA_{95}$, $OXA_{98}$, and $OXA_{116}$) (13) and the acquired $OXA_{23}$ (6 of 7 clinical isolates), and had AmpC ($ADC_{25}$). The analysis of the mutations in the septum formation PBP3 (proposed as the main FDC target) showed only a nonsynonymous mutation in Abau2, leading to the A515V amino acid change, and 2 nonsynonymous mutations in ACICU, producing A346V and H370Y amino acid variations. Moreover, we found 7 (ATCC 19606), 10 (all of the clinical strains), or 15 (ACICU) synonymous mutations in the same gene (reference genome *Acinetobacter baumannii* ATCC 17978 [GenBank accession number CP000521.1], locus tag A1S_3204). With regard to TonB3, a component of the iron uptake system and thus likely related to the FDC mechanism of entry into the cell, we found only 2 nonsynonymous mutations (H58R and A83E) in all of the clinical strains and ACICU, while ATCC 19606 had 1 more nonsynonymous mutation (V268A). Moreover, 8 synonymous mutations in all of the clinical strains and ACICU and 12 synonymous mutations in ATCC 19606 were discovered (reference genome *Acinetobacter baumannii* ATCC 17978 [GenBank accession number CP000521.1], locus tag A1S_3032). A complete list of single-nucleotide polymorphisms (SNPs) and related amino acid mutations is presented in Table S1 in the supplemental material.

**TABLE 1** STs, β-lactamase genes, and antibiotic resistance profiles of the study sample

| Strain | Genomic analysis result | | | MIC (mg/L) for[a]: | | MHA test result[b] | | MIC or MBC (geometric mean) in: | | | | PAP result | | |
| | ST | bla$_{OXA}$ | bla$_{ADC}$ | MEM | COL | DD diameter (mm) | S/HR/NS[c] | CAMHB MIC (mg/L) | ID-CAMHB MIC (mg/L) | MBC (mg/L) | MBC/MIC | 2 mg/L of FDC/antibiotic-free CFU/mL_proportion (%) | 32 mg/L of FDC/antibiotic-free CFU/mL ratio | S/HR/NS[d] |
|---|---|---|---|---|---|---|---|---|---|---|---|---|---|---|
| Abau1 | 2 | OXA$_{23,116}$ | ADC$_{25}$ | 64 (R) | 1 (S) | 12 | NS | 4 | 8 | 16 | 2 | >50 | >1/10$^6$ | HR |
| Abau2 | 2 | OXA$_{23,66}$ | ADC$_{25}$ | 64 (R) | 2 (S) | 24 | S | 1 | 0.5 | 2 | 4 | <50 | >1/10$^6$ | HR |
| Abau3 | 2 | OXA$_{23,116}$ | ADC$_{25}$ | 64 (R) | >64 (R) | 17 | S | 1 | 4 | 8 | 2 | >50 | >1/10$^6$ | HR |
| Abau4 | 52 | OXA$_{66,72}$ | ADC$_{25}$ | >64 (R) | 8 (R) | 16 | NS | 8 | 1 | 2 | 2 | >50 | >1/10$^6$ | NS |
| Abau5 | 2 | OXA$_{23,82}$ | ADC$_{25}$ | >64 (R) | 2 (S) | 15 i.c. | NS/HR | 4 | 4 | 4 | 1 | >50$^e$ | >1/10$^6$ | HR |
| Abau6 | 2 | OXA$_{23,82}$ | ADC$_{25}$ | >64 (R) | 1 (S) | 15 i.c. | NS/HR | 8 | 4 | 4 | 1 | >50 | >1/10$^6$ | HR |
| Abau7 | 2 | OXA$_{23,82}$ | ADC$_{25}$ | >64 (R) | 2 (S) | 15 | NS | 8 | 16 | 32 | 2 | >50 | >1/10$^6$ | HR |
| ACICU | 2 | OXA$_{20,66}$ | ADC$_{25}$ | 64 (R) | 1 (S) | 17 | S | 8 | 2 | 8 | 4 | >50 | >1/10$^6$ | HR |
| ATCC 19606 | 52 | OXA$_{98}$ | ADC$_{25}$ | 0.5 (S) | 1 (S) | 26 | S | 0.12 | 0.06 | 0.5 | 8 | <50$^e$ | >1/10$^6$ | HR |
| ATCC 17978 | 77 | OXA$_{95}$ | ADC$_{25}$ | 1 (S) | 1 (S) | 22 | S | 0.25 | 0.06 | 0.25 | 4 | <50$^e$ | <1/10$^6$ | S |

[a]MEM, meropenem; COL, colistin; S, susceptible; R, resistant.

[b]HR, heteroresistant; NS, nonsusceptible; i.c., colonies within the zone of clearing.

[c]Characterized as susceptible or nonsusceptible in accordance with the EUCAST breakpoints for FDC and AST. A zone diameter of ≥17 mm is typical for isolates with MIC values of ≤2 mg/L (*Enterobacterales, Pseudomonas aeruginosa*, and the pharmacokinetic/pharmacodynamic breakpoint) (19); the presence of colonies within a zone of clearing indicates a possible heteroresistance phenotype (14).

[d]Characterized as susceptible, heteroresistant, or nonsusceptible following the published protocol (6, 26).

[e]$P \leq 0.05$ (Student's t test).

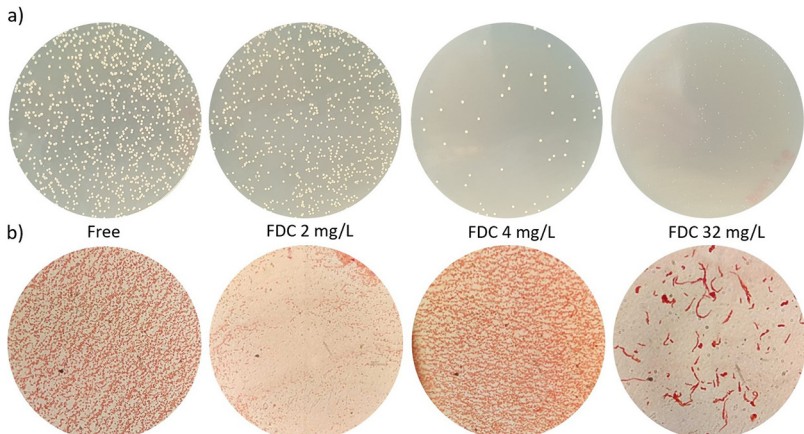

**FIG 1** Representative images of small colony growth (a) and microscopic shapes (b) of nonsusceptible or heteroresistant strains of *A. baumannii* at FDC concentrations higher than their MICs. The plates represent ACICU (FDC MIC, 2 mg/L) on TSA without antibiotic (Free) and in the presence of increasing concentrations of FDC. The same behavior was observed for all of the nonsusceptible and heteroresistant strains.

**MIC, MBC, and cidality.** Table 1 shows the meropenem and colistin MIC values together with the FDC *in vitro* susceptibility profile, in terms of MIC, MBC, and MBC/MIC ratio, to determine drug cidality. In accordance with the EUCAST criteria for susceptibility categorization, all strains except *A. baumannii* ATCC 19606 and ATCC 17978 were resistant to meropenem, while only 2 strains (namely, Abau3 and Abau4) were colistin resistant.

FDC DD results highlighted that 2 clinical *A. baumannii* strains and the 3 control strains were susceptible to FDC, while 4 clinical strains were nonsusceptible, with 2 strains showing colonies within the FDC inhibition zones, which itself demonstrates the presence of subpopulations, as reported by Sherman et al. (14).

MIC assays were performed in triplicate in both cation-adjusted Mueller-Hinton II broth (CAMHB) and iron-depleted CAMHB (ID-CAMHB), and their geometric means were compared with DD test results to evaluate concordance. Inhibition zone and MIC values were correlated for 7 strains, whereas 2 clinical strains (Abau3 and Abau4) showed discordance between DD and MIC results in ID-CAMHB, and the laboratory-adapted ACICU strain displayed different susceptibility phenotypes when tested by DD and in CAMHB for MIC evaluation. Of note, all discrepancies were observed when the inhibition zone was 17 mm (the susceptibility breakpoint) in size and the MIC values in ID-CAMHB were 1 dilution higher or lower than the suggested resistance breakpoint (2 mg/L). The MBC/MIC ratios evaluated with standard criteria (15) showed bactericidal activity for FDC in all strains except 1 (*A. baumannii* ATCC 19606).

**PAP analysis.** In addition to the previously described determinations, PAP analysis was performed for all of the study strains in order to determine the frequency of bacterial cells growing on agar supplemented with various concentrations of FDC. Results are reported in Table 1 and in Fig. S1 in the supplemental material. Eight of 10 strains were heteroresistant, as the numbers of colonies growing at antibiotic concentrations up to 16 times the breakpoint (that is, not just at 2 to 4 times, as stated in the guidelines) were at least $1/10^6$ of those growing on antibiotic-free plates. Abau5 was considered nonsusceptible because the colonies growing in the presence of 2 mg/L of FDC outnumbered 50% of the colonies growing on antibiotic-free plates, whereas *A. baumannii* ATCC 17978 was considered fully susceptible (see Fig. S2). Moreover, all strains except *A. baumannii* ATCC 17978 grew on Mueller-Hinton agar (MHA) plates with 32 mg/L of FDC in the form of very tiny colonies (Fig. 1a), and microscopic examinations of these colonies revealed the presence of filamentous bacteria (Fig. 1b). These colonies were slightly visible after 24 h of incubation and became big enough to be enumerated after 48 h. An internal control experiment revealed no differences in the colony growth on MHA plates with FDC that were incubated at 37°C for 48 h before inoculation and plates that were inoculated right after their preparation, suggesting FDC stability at 37°C for at least 2 days. Small colonies immediately turned back to their original shape once the drug was

**TABLE 2** Regrowth from MBC/MIC assays

| Strain | Regrowth (CFU/mL) with FDC at[a]: | | | | | | | | | |
|---|---|---|---|---|---|---|---|---|---|---|
| | 0.06 mg/L | 0.125 mg/L | 0.25 mg/L | 0.5 mg/L | 1 mg/L | 2 mg/L | 4 mg/L | 8 mg/L | 16 mg/L | 32 mg/L |
| Abau1 | un | un | un | un | un | un | un | un[b] | 90 | 0 |
| Abau2 | un | un | un | un[b] | un (sc) | un (sc) | 0 | 0 | 0 | 0 |
| Abau3 | un | un | un | un | un | un | un[b] | 0 | 0 | 3,000 |
| Abau4 | un | un | un | un | un[b] | 0 | 0 | 0 | 0 | 0 |
| Abau5 | un | un | un | un | un | un | 100[b] | 0 | 1,000 | un |
| Abau6 | un | un | un | un | un | un | 0[b] | 0 | 0 | 0 |
| Abau7 | un | un | un | un | un | un | un | un | un[b] | 0 |
| ACICU | un | un | un | un | un | un[b] | un | 0 | 0 | 1,800 |
| ATCC 19606 | un[b] | un | 340 | 230 | 0 | 0 | 0 | 0 | un | 9,400 |
| ATCC 17978 | un[b] | un | 0 | 0 | 0 | 0 | 0 | 0 | 0 | 0 |

[a]Values are the means of two different experiments. un, uncountable; sc, small colonies.
[b]MIC value for the strain.

removed (data not shown). These data confirm the use of standard laboratory testing to detect heteroresistance, which was very widespread in our isolates.

**Regrowth and stability of nonsusceptible induced isolates.** During MBC experiments, 4 strains showed regrowth of approximately $10^3$ CFU/mL at the highest FDC concentration of 32 mg/L, despite proving susceptible to FDC by DD testing and the drug demonstrating cidal activity (Table 2). Among the strains showing regrowth after MIC and MBC assays, Abau3 was the only one able to grow after various passages on MHA plates and in ID-CAMHB containing 32 mg/L of FDC. As expected, the FDC MIC value of the strain picked up from an MHA plate containing FDC was significantly higher than the original MIC for Abau3, i.e., MIC values of >32 mg/L versus 4 mg/L. Nevertheless, nonsusceptibility was lost after 2 passages of the induced Abau3 strain on antibiotic-free plates (Table 3). A DD test with induced Abau3 gave an inhibition zone of 6 mm, confirming its nonsusceptible profile, while the same test performed with Abau3 that had lost induction yielded an inhibition zone of 18 mm, as typical for susceptible strains (Table 3). Microscopic investigations of induced Abau3 showed no or very few differences in the bacterial cells grown in the presence or absence of FDC, with only a few filamentous bacteria under the former conditions (Fig. 2).

**Effects of $\beta$-lactamase inhibitors on FDC susceptibility restoration.** Because our strains harbored multiple classes of $\beta$-lactamase resistance genes, the addition of $\beta$-lactamase inhibitors (BLIs) to FDC could to some extent restore its activity. The inhibition zones of ceftazidime and ceftazidime-avibactam for the nonsusceptible induced Abau3 strain grown on antibiotic-free MHA plates and plates with 32 mg/L of FDC are shown in Fig. 3. The lack of inhibition around the disks containing ceftazidime and ceftazidime-avibactam on antibiotic-free MHA plates indicated that the strain was resistant to ceftazidime and that avibactam would not increase its efficacy against the resistant strain. Conversely, the emergence of an inhibition zone around the ceftazidime-avibactam disk, together with its absence around the ceftazidime disk, on MHA plates supplemented with FDC revealed that avibactam was able to restore FDC antimicrobial activity. Ampicillin-sulbactam and ampicillin strips did not show any difference in plates with or without FDC (data not shown), indicating poor activity of sulbactam in restoring the efficacy of FDC in this induced isolate.

The same tests performed with the other strains with FDC MICs of >1 mg/L (i.e., Abau1, Abau3, Abau5, Abau6, Abau7, and ACICU) revealed different activities for avibactam and sulbactam regardless of their FDC MIC values. Particularly, avibactam restored FDC activity for

**TABLE 3** Induction and maintenance of FDC nonsusceptibility

| Strain | MHA test result | | MIC (mg/L) in ID-CAMHB |
|---|---|---|---|
| | DD (mm) | S/NS[a] | |
| Abau3 postinduction with 32 mg/L from agar plate with 32 mg/L FDC | 6 | NS | >32 |
| Abau3 postinduction with 32 mg/L from FDC-free agar plate | 18 | S | 2 |

[a]Characterized as susceptible (S) or nonsusceptible (NS) in accordance with the EUCAST breakpoints for FDC and AST; a zone diameter of ≥17 mm is typical for isolates with MIC values of ≤2 mg/L (*Enterobacterales*, *Pseudomonas aeruginosa*, and the pharmacokinetic/pharmacodynamic breakpoint) (31).

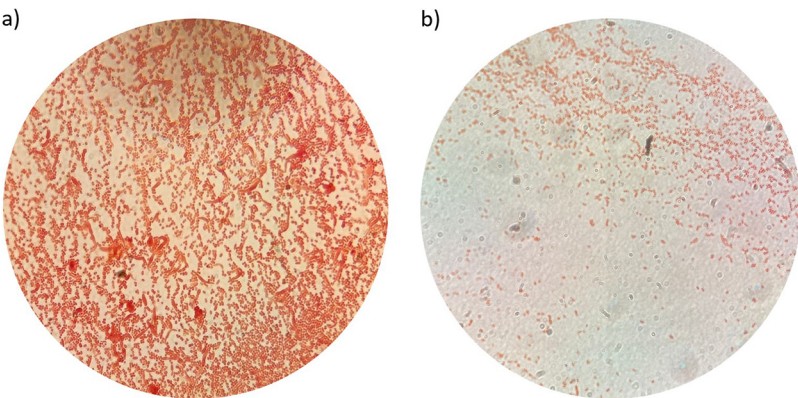

**FIG 2** Microscopic snapshots of induced Abau3 in the presence and in the absence of FDC.

Abau3 and Abau7, while sulbactam restored FDC activity for the Abau3, Abau5, Abau6, and Abau7 strains (resistant to ampicillin-sulbactam with MIC values of >256 mg/L) (see Table S2).

## DISCUSSION

FDC gained great attention due to its peculiar mechanism of entry into cells (the so-called Trojan horse-like approach [4]) and undoubted activity against multidrug-resistant (MDR) organisms, including *A. baumannii* (3, 16, 17). In this context and in parallel with the increased expectations of its use, difficulties emerged in terms of (i) the definition of the best method to be used for AST, resulting from the peculiar requirement for the drug to be assessed in ID-CAMHB in order to induce siderophore-mediated entry, and (ii) some cases of clinical and microbiological failures reported in the literature, related to evoked heteroresistance to this drug (5, 18). From what is currently known, major gaps in all of these fields are still to be filled.

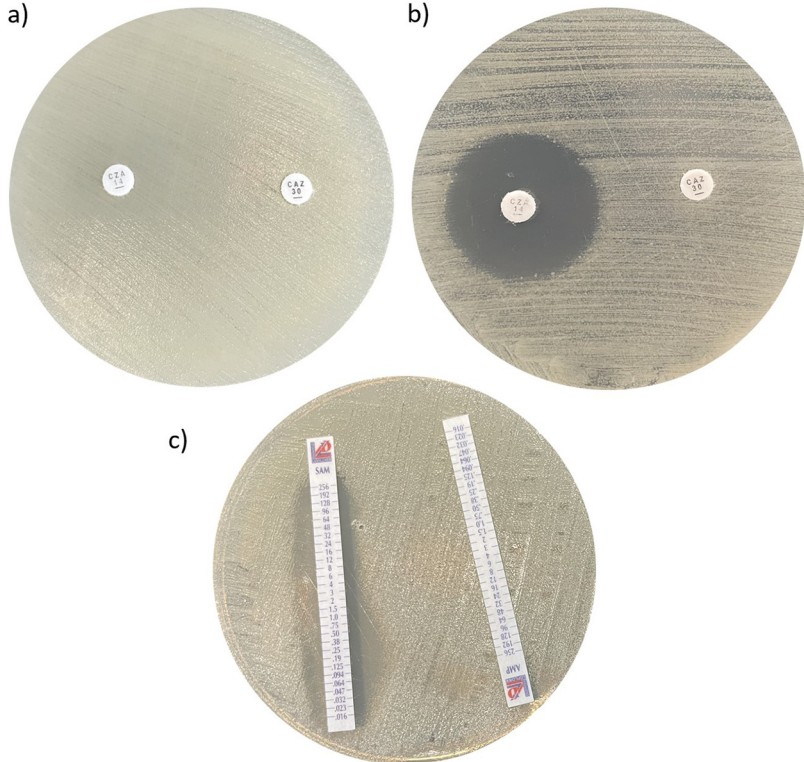

**FIG 3** Effects of BLIs on FDC susceptibility restoration.

Our study aimed to test FDC susceptibility in some strains of *A. baumannii*, both clinical and reference strains, focusing initially on the conditions for AST with the use of two different liquid media and the DD method to determine the level of reproducibility of each testing set; further, we aimed to determine the cidality of the drug, followed by PAP analysis experiments to evaluate the presence of resistant cell subpopulations, as well as the activity of BLIs in restoring the antibacterial activity of FDC.

The *A. baumannii* strains included in the study belonged to three different STs and were MDR, with resistance to meropenem (8/10 isolates) and, to a lesser extent, colistin (2/10 isolates). Resistome analysis by NGS demonstrated a plethora of OXA variants and the presence of an ADC gene in their genomes. In these strains, FDC showed different degrees of susceptibility when a DD breakpoint of ≥17 mm was used (5 strains were susceptible and 5 were nonsusceptible or heteroresistant due to the presence of colonies within the inhibition zone). The geometric mean of the MIC values determined at least three times in CAMHB and ID-CAMHB correlated better with the DD results when the DD and MIC values were distant from their respective breakpoints (17 mm and 2 mg/L, respectively); in contrast, when those values coincided or were very close to the breakpoints, the rate of discordance between the two methods was higher. The same discordance can be found in the EUCAST MIC-zone diameter correlates for *A. baumannii* for strains with DD diameters between 16 and 18 mm (area of technical uncertainty [ATU]) (19).

Considering the MBC/MIC ratio, even if FDC proved bactericidal in almost all isolates, *A. baumannii* seemed intrinsically prone to adapt to increased drug concentrations, showing a general ability (4 of 10 strains) to survive at the highest drug concentration. In particular, these subpopulations exhibited smaller colonies and a change in the shape of the microorganisms, from a coccobacillary form to filamentous cells. The regrowth capacity of several tiny colonies (as demonstrated at the highest concentrations of FDC) retested immediately after isolation and after 2 passages in antibiotic-free medium, was found to be unstable. Indeed, all colonies reverted to the original MIC value. The change in the shape of the microorganisms could be explained by the mechanism of inhibition of FDC, which binds PBP3 as the most important target of the drug, followed by PBP2 and PBP1 (20), as well as by the propensity of *A. baumannii* to change its colony morphology and cell shape in response to many stress factors (21). However, the different resistance profiles of our strains seem to be not related to PBP3 integrity, as there were no amino acid changes in 7 samples, compared to the fully susceptible ATCC 17978 strain, and only 1 or 2 nonsynonymous mutations in the remaining 2 isolates. Of note, the H370Y amino acid change found in ACICU was reported previously by Nordmann et al. for an *A. baumannii* strain following treatment with FDC that resulted in an increase of the MIC value from 1 to 4 mg/L (22). Of note, the regrowth observed during MBC/MIC assays was more evident at the highest FDC concentration and, in some cases, there was no colony growth at FDC concentrations between the MIC values (indicated in Table 2) and 16 to 32 mg/L. The same behavior was observed during PAP experiments. In our opinion and due to the lack of scientific evidence, it may correlate with the activation of regulatory circuitries promoted by the higher antibiotic concentrations.

Genomic analyses also revealed the presence of mutations in the *ton*B3 gene, whose product (TonB3) is part of the system that provides Gram-negative bacteria with the energy needed to transport host iron-carrier and iron-siderophore complexes into the periplasm once these complexes are bound to cognate TonB-dependent outer membrane receptors (23) and thus may be related to the entry of FDC into bacterial cells. The 2 mutations found were shared by all of the clinical strains as well as ACICU and ATCC 19606, with the latter strain having 1 additional mutation. Mutations were present in all of the strains regardless of the resistance phenotype, giving us the idea that it is unlikely that they could be associated with the FDC resistance. Moreover, it is well known that the TonB3-dependent system of iron transport is not the only way for FDC to enter the cell (24).

Regrowth at the highest concentration and its loss after as few as 2 passages without FDC confirm the adaptability of this species to the surrounding environment, responding to a myriad of intracellular and extracellular signals (25). All of these regulatory circuitries, which are widely represented in the genome of this microorganism, were evoked to explain

the emergence of resistant cells expressing distinct and critical phenotypes (for example, becoming resistant) in order to survive and adapt to hostile situations.

The same phenomenon of regrowth at increased FDC concentrations in populations of $>1 \times 10^6$ CFU/mL was observed during PAP. Of the 10 strains tested, only 1 was frankly susceptible and 1 frankly nonsusceptible; all others could be defined as heteroresistant to the drug. Even in this case, tiny colonies and morphological changes were observed. In our experiments, very tiny colonies started to grow right after the first overnight incubation but were too small to be counted, and they continued to grow larger up to 48 h of incubation, to reach a size that allowed us to count them and pick them up. Once again, our *A. baumannii* FDC susceptibility profiles were inconsistent when PAP results were analyzed. In fact, only the frankly nonsusceptible and frankly susceptible strains were classified in the same way based on DD and MIC evaluations. Despite being quick and easy to perform, these latter methods are not suited to determining the presence of unstable heteroresistant subpopulations, which could be the reason (suggested but not confirmed) for some therapeutic failures with FDC and other antibiotics (5, 6, 18). These cases need to be further investigated by PAP to better classify *A. baumannii* resistance profiles, as suggested by Band et al. for *Enterobacterales* (26).

Fortunately, the combination of FDC with a BLI, such as avibactam or sulbactam, seems to restore FDC antibacterial activity even at concentrations several times lower than its MIC, as already shown by other authors (27). It remains unclear why the synergistic activity was provided by both of the tested BLIs in some strains while only one inhibitor restored the activity of FDC in other strains. Furthermore, no correlation was evident between FDC MIC values and the activity of BLIs or the presence of different $bla_{OXA}$ genes.

**Conclusions.** The need for new antimicrobial agents to address the threat of antibiotic resistance is recognized worldwide and, in this scenario, new molecules such as FDC are a breath of fresh air. However, any new antibiotics need to be thoroughly investigated in order to characterize their strengths and weakness and, perhaps more importantly, to determine antibiotic-bacterial species interconnections.

FDC demonstrated promising activity against Gram-negative bacteria, including those referred to as ESKAPE pathogens (such as *A. baumannii*), for which the therapeutic options are limited. By investigating the FDC resistance profiles of some clinical and laboratory-adapted *A. baumannii* strains, we found that these strains are not easily classified into the commonly used categories of susceptibility, especially strains with susceptibility levels very close to the proposed resistance breakpoints, due to a high prevalence of heteroresistant subpopulations. Heteroresistance in *A. baumannii* seems to be common as a stress response mechanism, but fortunately it is apparently an unstable and transient trait. Our results, in accordance with recently published clinical observations (12, 28), support the use of combination therapy when FDC is used to treat severe *A. baumannii* infections.

## MATERIALS AND METHODS

**Bacterial strains and culture conditions.** Three laboratory-adapted and 7 clinical MDR *A. baumannii* strains were selected for this study. Laboratory-adapted *A. baumannii* ATCC 19606 and ATCC 17978 were commercially obtained from the American Type Culture Collection (ATCC) (Manassas, VA), and *A. baumannii* ACICU was provided by Paolo Visca (Roma Tre University, Italy), while clinically significant strains were recently isolated from bronchial aspirate samples from FDC-naive hospitalized patients, with FDC testing requested by the infectious disease specialist. The clinical isolates were identified as *A. baumannii* by matrix-assisted laser desorption ionization–time of flight (MALDI-TOF) mass spectrometry (Bruker Daltonics, Billerica, MA, USA). The strains were grown on McConkey agar plates (Oxoid, UK) at 37°C for 18 h and stored in Trypticase soy broth (TSB) (Oxoid) with 15% glycerol at −80°C until further analysis.

**Genome extraction, sequencing, and analyses. (i) DNA extraction.** DNA extraction was carried out with the QIAamp DNA minikit (product 51304; Qiagen, Hilden, Germany) following the manufacturer's protocol. DNA was quantified using an Eppendorf BioPhotometer D30 and the Qubit fluorimeter double-stranded DNA (dsDNA) broad-range (BR) assay kit (product 32850; Invitrogen, Carlsbad, CA, USA).

**(ii) NGS.** Whole-genome sequencing was performed on a MiSeq platform according to the manufacturer's instructions provided in the Illumina DNA preparation (M)-tagmentation (24 samples) for Illumina kit (product 20018707; Illumina, Inc., San Diego, CA, USA) using 100 ng of extracted DNA. Indexes were provided with the Nextera DNA CD indexes (24 indexes and 24 samples) for Illumina kit (product 20019105; Illumina).

Denature and dilute libraries were prepared by following the protocol provided by Illumina (29) and choosing a loading concentration of 8.5 pM. Finally, sequencing was performed using the MiSeq reagent kit v3

(product 15043895; Illumina). The creation of the sample sheet was accomplished by using Local Run Manager v3 software and following the instructions in the Local Run Manager v3 software guide provided by Illumina (30).

**(iii) Bioinformatic analysis.** ACICU (GenBank accession number NC_010611.1) and ATCC 19606 (GenBank accession number AP022836) fasta and GenBank files were downloaded from the National Center for Biotechnology Information (NCBI) (31). The paired-end raw reads for Abau1, Abau2, Abau3, Abau4, Abau5, Abau6, and Abau7 were analyzed with BacPipe (32), with built-in parameters, using as a reference *Acinetobacter baumannii* ATCC 17978 (GenBank accession number CP000521.1). Moreover, to identify point mutations, the output assemblies from BacPipe for each sample were aligned, using as references two protein sequences, PBP3 (GenBank accession number ABO13597.2) and TonB3 (GenBank accession number ABO13429.2), through the Burrows-Wheeler Aligner (BWA) (33). Finally, files were sorted using SAMtools, variants were called, and consensus sequences were generated using BCFtools (34, 35). Differences in amino acid sequences were detected using BLASTp (36).

**AST.** *In vitro* susceptibility to meropenem, colistin, and FDC was evaluated in accordance with the EUCAST guidelines (19, 37). Meropenem trihydrate and colistin sulfate salt powders (Merck/Sigma-Aldrich, Germany) were used to assess MICs by BMD in CAMHB (Becton, Dickinson and Company, MD, USA).

FDC disks containing 30 $\mu$g of antibiotic (Liofilchem, Italy) and FDC powder (Shionogi & Co., Ltd., Osaka, Japan) were provided by Shionogi. DD assays were performed on MHA plates (Oxoid), while BMD MIC determinations were performed with both CAMHB and ID-CAMHB, the latter as indicated in the EUCAST guidelines, prepared using Chelex 100 resin (Bio-Rad Laboratories, Hercules, CA, USA) as reported by Hackel et al. (11). All experiments were performed in triplicate, and the geometric mean was then calculated.

Due to the lack of actual breakpoints for FDC susceptibility in *A. baumannii*, strains with inhibition zones with a diameter of $\geq$17 mm were considered susceptible, whereas a diameter of <17 mm was indicative of nonsusceptible *A. baumannii* strains. Similarly, strains with a MIC value of >2 mg/L were considered nonsusceptible to FDC (37). The MBC was determined by plating the dilution representing the MIC, as well as 2 less concentrated dilutions and 5 more concentrated dilutions from the same MIC assay, on MHA plates and counting viable colonies. MBC was defined as the lowest concentration causing at least a 99.9% reduction in CFU per milliliter, compared to the starting inoculum. The MBC/MIC ratio was calculated from MIC assays performed in ID-CAMHB to determine whether FDC had bactericidal or bacteriostatic activity with *A. baumannii* strains; a ratio of $\leq$4 was indicative of a bactericidal effect (15).

**PAPs.** PAPs were determined and evaluated as described previously by Choby et al. and Band et al. (6, 26) with minor modifications. Briefly, a well-isolated colony from a McConkey agar plate was inoculated in 3 mL of TSB (Oxoid) and incubated overnight at 37°C with shaking. The bacterial culture was serially diluted in sterile H$_2$O from $10^{-1}$ to $10^{-6}$, and 50 $\mu$L of each dilution was plated in duplicate on MHA plates containing 0 (antibiotic-free), 1, 2, 4, 8, 16, or 32 mg/L of FDC (limit of quantification, 20 CFU/mL). Colonies were enumerated after 24 and 48 h of growth at 37°C, and only plates with 10 to 300 colonies were considered for subsequent analysis. Isolates were classified as nonsusceptible if the number of colonies that grew at the breakpoint concentration (i.e., 2 mg/L) was $\geq$50% of the number growing on antibiotic-free plates. If an isolate appeared susceptible, then it was classified as heteroresistant if the number of colonies that grew at the most concentrated dilution of FDC (i.e., 32 mg/L) was at least 1/10$^6$ of the number growing on antibiotic-free plates. Isolates that were neither nonsusceptible nor heteroresistant were classified as susceptible. Student's *t* tests of the ratios between the number of CFU per milliliter grown on MHA plates with 2 mg/L of FDC and 50% of the colonies grown on FDC-free plates were performed using GraphPad Prism v8.0.0 for MacOS (GraphPad Software, San Diego, CA, USA), and only *P* values of $\leq$0.05 were considered statistically significant. The morphology of the bacteria exposed to different concentration of FDC was evaluated by optical microscopy.

**FDC nonsusceptibility induction.** Starting from the PAP assay, 1 colony was recovered from an MHA plate with 32 mg/L of FDC, streaked on a fresh MHA plate with 32 mg/L of FDC, and inoculated in 3 mL of ID-CAMHB with 32 mg/L of FDC. If the strain was able to grow in the presence of such a high concentration of antibiotic, then a DD test and MIC evaluation were performed as described above, to evaluate a potential change in the susceptibility profile of the strain. The same tests were repeated after culturing of the previously induced strain on FDC-free MHA plates for 2 days, to determine whether the acquired nonsusceptibility to the drug was stable. The morphology of the induced bacteria, either exposed or not to the highest concentration of FDC, was evaluated by optical microscopy.

**Evaluation of the synergistic activity of avibactam and sulbactam.** The synergistic activity of the two BLIs in combination with FDC was investigated by DD test or Etest using disks of ceftazidime and ceftazidime-avibactam (Oxoid) containing 10 $\mu$g of ceftazidime and 4 $\mu$g of avibactam or strips of ampicillin and ampicillin-sulbactam (Liofilchem, Italy) containing 0.016 to 256 mg/L of ampicillin and 0.008 to 128 mg/L of sulbactam, placed on both FDC-free MHA plates and MHA plates containing 32 mg/L or 1 mg/L of FDC that had been previously inoculated with an FDC-nonsusceptible induced strain (FDC MIC of >32 mg/L) or strains with FDC MICs of >1 mg/L. *A. baumannii* ATCC 17978 was used as the activity control with 1 mg/L of FDC. The comparison between the inhibition obtained with ceftazidime and ceftazidime-avibactam or ampicillin and ampicillin-sulbactam in the plates with and without FDC allowed us to evaluate the efficacy of the BLIs.

**Data availability.** Clinical strain sequences are available at the SRA under BioProject accession number PRJNA850903.

## SUPPLEMENTAL MATERIAL

Supplemental material is available online only.
**SUPPLEMENTAL FILE 1**, PDF file, 0.3 MB.

## ACKNOWLEDGMENTS

We thank Shionogi & Co., Ltd. for supplying FDC powder, disks, and strips. We thank Paolo Visca for providing the ACICU strain. We also thank PharmaTranslated (http://www .pharmatranslated.com) and particularly Silvia Montanari for the language revision.

The work was partially supported by a research grant (grant PRIN2020) from the Ministry of Research (MIUR), Italy, and was partially funded by Shionogi & Co., Ltd.

S. Stracquadanio and S. Stefani conceptualized the work and wrote the manuscript, S. Stracquadanio, C. Bonomo, A. Marino, and A. Mirabile performed the experiments, D. Bongiorno, G. F. Privitera, D. A. Bivona, and P. G. Bonacci performed genome sequencing and mutational analysis, S. Stracquadanio, C. Bonomo, and S. Stefani analyzed the data, and S. Stefani supervised and acquired the funds. All authors have read and agreed to the published version of the manuscript.

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
