## [Reviewer comments · Microbiology Spectrum]

Microbiology Spectrum

Acinetobacter baumannii and cefiderocol between cidality and adaptability

Stefano Stracquadanio, Carmelo Bonomo, Andrea Marino, Dafne Bongiorno, Grete Privitera, Dalida Bivona, Alessia Mirabile, Paolo Bonacci, and Stefania Stefani

Corresponding Author(s): Stefania Stefani, University of Catania, Catania, Italy

Review Timeline:

Submission Date:	June 22, 2022
Editorial Decision:	August 13, 2022
Revision Received:	August 31, 2022
Accepted:	September 7, 2022

Editor: Paolo Visca

Reviewer(s): The reviewers have opted to remain anonymous.

Transaction Report:

DOI: <https://doi.org/10.1128/spectrum.02347-22>

August 13, 2022

Prof. Stefania Stefani
University of Catania, Catania, Italy
Catania
Italy

Re: Spectrum02347-22 (Acinetobacter baumannii and cefiderocol between cidalty and adaptability)

Dear Prof. Stefania Stefani, cara Stefania:

Thank you for submitting your manuscript to Microbiology Spectrum. I have read with great interest your manuscript, and I feel it provides relevant information on cefiderocol activity against *A. baumannii*. Overall, the manuscript is well written, but I found some points which need to be better explained or strengthened. I hope your manuscript will benefit from my suggestions (see below).

When submitting the revised version of your paper, please provide (1) point-by-point responses to the issues I have raised (see below) as file type "Response to Reviewers," not in your cover letter, and (2) a PDF file that indicates the changes from the original submission (by highlighting or underlining the changes) as file type "Marked Up Manuscript - For Review Only". Please use this link to submit your revised manuscript - we strongly recommend that you submit your paper within the next 60 days or reach out to me. Detailed instructions on submitting your revised paper are below.

Link Not Available

EDITOR'S COMMENTS

L. 30, 129,209, 338, 339; Please notice that *A. baumannii* has 3 tonB orthologues (named tonB1, tonB2, and tonB3 see doi: 10.1128/IAI.00540-13) but only tonB3 is essential for siderophore-mediated iron uptake (see doi: 10.1128/IAI.00755-18). Authors should specify in which of the three TonB orthologous proteins the mutations were detected (indicate the reference sequence for blastp comparisons in Materials and Methods).

L. 66: Please rephrase "Heteroresistance has been suggested as a possible reason for these failures....." (please notice that that "These authors..." is inappropriate since no Author is cited).

L. 84: Given the elusive nature of heteroresistance, I am not convinced that the term "induction" is appropriate in this part of the text (as you write "..... which in some cases are the result of antibiotic induction....."). I suggest to rephrase as follows; "..... which in some cases emerged upon exposure to FDC". Later in the text, after having explained the induction experiment, the term "induction" seems appropriate.

L. 125: Please spell NCBI

L. 129: Please spell BWA

L. 131: Please rephrase "Differences in amino acid sequences were detected using blastp....."

L. 152: Please rephrase "causing 99.9% reduction"

L. 169: Please rephrase "Isolates that were neither non-susceptible nor heteroresistant were classified as susceptible."

L. 201: Please delete "n." and spell "ST 2 (8 strains), ST 52 (1 strain), ST 77 (1 strain)."

Table 1: Strain ATCC 19606 has been assigned to the ST52 by multi-locus sequence typing (MLST) [see <https://doi.org/10.1016/j.ijmm.2020.151412>]. Please correct accordingly or verify your assignment again.

Table 1 and L. 202: All *A. baumannii* strains carry the chromosomal *oxa51* carbapenemase gene, commonly used as a genetic marker of the species (also included in the MLST scheme). This resistance gene should be included in the Table.

L. 202, 219 and 242: Table 1 only (I can't see sections, a, b in the revised Table 1).

L. 215: Please spell "single-nucleotide polymorphisms (SNPs)"

L. 248 and 250; Sections A and B of Fig. 1 should be labelled on the figure. A more descriptive legend would help the reader.

Table 2 reads with difficulty; please indicate the meaning of boxed values in footnotes. Moreover, it is impressive to see that many strains yield colonies on 32 mg/L FDC but no colonies (zero value) on much lower concentrations of the drug (2-16 mg/L). Unless I am missing some relevant aspects of the experiment, this result deserves more in depth discussion and better clarification along the text.

L. 284-5: It is reported that SAM and AMP strips did not show any difference in plates with or without FDC, however Fig. 3 shows only a plate with FDC 1 mg/L and a clear inhibition zone in the e-test with SAM. Authors conclude that SAM has a poor activity in restoring the efficacy of FDC in the induced isolate. This conclusion should be corroborated by illustrations of the result obtained with SAM on FDC-free plates.

L. 293: Please rephrase "Cefiderocol (FDC), a novel siderophore cephalosporin, gained" simply as "FDC gained" (not so novel drug nowadays).

L. 309: Please rephrase "(8/10 isolates).....(2/10 isolates)". Don't use sample.

Sincerely,

Paolo Visca

Journals Department
Reviewer comments:

Staff Comments:

Preparing Revision Guidelines

Please return the manuscript within 60 days; if you cannot complete the modification within this time period, please contact me. If you do not wish to modify the manuscript and prefer to submit it to another journal, please notify me of your decision immediately so that the manuscript may be formally withdrawn from consideration by Microbiology Spectrum.

Dear Paolo Visca,

we are grateful for the interest you expressed about our work and we really appreciate all your comments. After the improvements you asked us to make to the manuscript, we now hope that the updated version of our paper will meet the standard of quality needed for the publication on Microbiology Spectrum.

Moving to the point-by-point responses to your concerns:

L. 30, 129,209, 338, 339; Please notice that *A. baumannii* has 3 tonB orthologues (named tonB1, tonB2, and tonB3 see doi: 10.1128/IAI.00540-13) but only tonB3 is essential for siderophore-mediated iron uptake (see doi: 10.1128/IAI.00755-18). Authors should specify in which of the three TonB orthologous proteins the mutations were detected (indicate the reference sequence for blastp comparisons in Materials and Methods).

We checked the sequences and our alignment has been done versus TonB3. We corrected all the TonB in the text and added the reference sequences of pbp3 and tonb3 (lines 127-128)

L. 66: Please rephrase "Heteroresistance has been suggested as a possible reason for these failures....." (please notice that that "These authors..." is inappropriate since no Author is cited).

Done

L. 84: Given the elusive nature of heteroresistance, I am not convinced that the term "induction" is appropriate in this part of the text (as you write "..... which in some cases are the result of antibiotic induction....."). I suggest to rephrase as follows; "..... which in some cases emerged upon exposure to FDC" . Later in the text, after having explained the induction experiment, the term "induction" seems appropriate.

Done

L. 125: Please spell NCBI

Done

L. 129: Please spell BWA

Done

L. 131: Please rephrase "Differences in amino acid sequences were detected using blastp....."

Done

L. 152: Please rephrase "causing 99.9% reduction"

Done

L. 169: Please rephrase "Isolates that were neither non-susceptible nor heteroresistant were classified as susceptible."

Done

L. 201: Please delete "n." and spell "ST 2 (8 strains), ST 52 (1 strain), ST 77 (1 strain)."

Done, and we modified the numbers of ST2 and ST52 strains as ATCC 19606 was now correctly assigned to ST52.

Table 1: Strain ATCC 19606 has been assigned to the ST52 by multi-locus sequence typing (MLST) [see <https://doi.org/10.1016/j.ijmm.2020.151412>]. Please correct accordingly or verify your assignment again. We performed the MLST again and the result corresponds to what is reported in literature. We modified the table and the ST list in the text (see previous comment).

Table 1 and L. 202: All *A. baumannii* strains carry the chromosomal *oxa51* carbapenemase gene, commonly used as a genetic marker of the species (also included in the MLST scheme). This resistance gene should be included in the Table.

According to literature analyses, OXA66, OXA82, OXA95, OXA98, and OXA116 (harbored by our strains) are OXA51-like enzymes. Unfortunately, there is still confusion in the *oxa* gene nomenclature and even the presence of a single SNP results in a new *oxa* number. We added a sentence in the text (lines 202-203) and a reference [24], specifying that all these *oxa* are of the same group of OXA51.

L. 202, 219 and 242: Table 1 only (I can't see sections, a, b in the revised Table 1).

a) and b) are reported in the first line of the table and we added a new vertical edge in the table to better separate the two sections.

L. 215: Please spell "single-nucleotide polymorphisms (SNPs)"

Done.

L. 248 and 250; Sections A and B of Fig. 1 should be labelled on the figure. A more descriptive legend would help the reader.

Done. We also added a) and b) in the figure caption.

Table 2 reads with difficulty; please indicate the meaning of boxed values in footnotes. Moreover, it is impressive to see that many strains yield colonies on 32 mg/L FDC but no colonies (zero value) on much lower concentrations of the drug (2-16 mg/L). Unless I am missing some relevant aspects of the experiment, this result deserves more in depth discussion and better clarification along the text.

We added the sentence "boxed squares indicate MIC values" in the footnotes and discussed the results at lines 336-340. This behavior was surprising for us too, but both MBC/MIC and PAP experiments showed a higher number of colonies at a concentration of cefiderocol equal to 16-32 mg/L. We can just hypothesize something similar to the paradox-effect due to the activation of regulatory circuitries activate by the drug high concentration.

L. 284-5: It is reported that SAM and AMP strips did not show any difference in plates with or without FDC, however Fig. 3 shows only a plate with FDC 1 mg/L and a clear inhibition zone in the e-test with SAM.

Authors conclude that SAM has a poor activity in restoring the efficacy of FDC in the induced isolate. This conclusion should be corroborated by illustrations of the result obtained with SAM on FDC-free plates.

Figure 3 shows an example of the behavior of Abau3, Abau5, Abau6 and Abau7 strains for which the sulbactam restores cefiderocol activity. The lack of activity is intended for the induced strain (induced Abau3) for which no clear zones were visible both with SAM and AMP in the presence or in the absence of

cefiderocol. Lines 278-286 and table 3 refer only to the Abau3-induced strain, whilst lines 287-291 and figure 3 describes the behavior of all the other native resistant strains.

L. 293: Please rephrase "Cefiderocol (FDC), a novel siderophore cephalosporin, gained" simply as "FDC gained" (not so novel drug nowadays).

Done.

L. 309: Please rephrase "(8/10 isolates).....(2/10 isolates)". Don't use sample.

Done.

You will find all our modifications highlighted in yellow in the PDF file.

Once again, thank you for all your suggestions,

Best Regards

Stefania Stefani

September 7, 2022

Prof. Stefania Stefani
University of Catania, Catania, Italy
Catania
Italy

Re: Spectrum02347-22R1 (Acinetobacter baumannii and cefiderocol between cidalty and adaptability)

Dear Prof. Stefani, cara Stefania:

Your manuscript has been accepted, and I am forwarding it to the ASM Journals Department for publication. You will be notified when your proofs are ready to be viewed.

Sincerely,

Paolo Visca
Editor, Microbiology Spectrum
